# Early steps of protein disaggregation by Hsp70 chaperone and class B J-domain proteins are shaped by Hsp110

Wiktoria Sztangierska, Hubert Wyszkowski, Maria Pokornowska, Klaudia Kochanowicz, Michal Rychłowski, Krzysztof Liberek*, Agnieszka Kłosowska*

Intercollegiate Faculty of Biotechnology of University of Gdańsk and Medical University of Gdańsk, University of Gdańsk, Gdańsk, Poland

*For correspondence:
krzysztof.liberek@ug.edu.pl (KL);
agnieszka.klosowska@ug.edu.pl (AK)

Competing interest: The authors declare that no competing interests exist.

**Abstract** Hsp70 is a key cellular system counteracting protein misfolding and aggregation, associated with stress, ageing, and disease. Hsp70 solubilises aggregates and aids protein refolding through substrate binding and release cycles regulated by co-chaperones: J-domain proteins (JDPs) and nucleotide exchange factors (NEFs). Here, we elucidate the collaborative impact of Hsp110 NEFs and different JDP classes throughout Hsp70-dependent aggregate processing. We show that Hsp110 plays a major role at initial stages of disaggregation, determining its final efficacy. The NEF catalyses the recruitment of thick Hsp70 assemblies onto aggregate surface, which modifies aggregates into smaller species more readily processed by chaperones. Hsp70 stimulation by Hsp110 is much stronger with class B than class A JDPs and requires the auxiliary interaction between class B JDP and the Hsp70 EEVD motif. Furthermore, we demonstrate for the first time that Hsp110 disrupts the JDP-Hsp70 interaction. Such destabilisation of chaperone complexes at the aggregate surface might improve disaggregation, but also lead to the inhibition above the sub-stoichiometric Hsp110 optimum. Thus, balanced interplay between the co-chaperones and Hsp70 is critical to unlock its disaggregating potential.

## eLife assessment

This study provides an **important** insight into the mechanisms of cooperation between Hsp70 and its cochaperones during reactivation of aggregated proteins. Based on **convincing** evidence, the authors demonstrate that the co-chaperone Hsp110 boosts disaggregation activity by enhancing Hsp70 recruitment to protein aggregates. This work is of broad interest to biochemists and cell biologists working in the protein homeostasis field.

## Introduction

During stress, protein homeostasis is perturbed by protein misfolding and aggregation. Accumulation of protein aggregates disrupts cellular functions and contributes to ageing and is a hallmark of many neurodegenerative diseases (*Hartl et al., 2011*; *Morimoto, 2008*). A protein quality control system, relying on a network of cooperating chaperones, has evolved to moderate stress-induced proteotoxicity and rebalance proteostasis.

To refold proteins from aggregates into their native state, eukaryotes use a system comprising Hsp70, an ATP-dependent chaperone, and its co-chaperones: an Hsp110 nucleotide exchange factor (NEF) and a J-domain protein (JDP/Hsp40), together with an Hsp100 disaggregase (*Glover and Lindquist, 1998*). The process is initiated by a JDP, which delivers Hsp70 to an aggregated substrate and

**eLife digest** For proteins to accurately carry out their role in the cell, they must first be precisely folded into specific 3D shapes. Stress, aging or disease can interfere with this delicate process, leading to misfolded proteins clumping together and causing damage. In response, the cell can deploy 'chaperones' which disentangle these aggregates and ensure that proteins recover their proper structure. Chaperones from the Hsp70 protein family, for example, are crucial for cell survival, especially under biologically stressful conditions. Yet Hsp70 proteins cannot perform their role without the assistance of co-chaperones such as Hsp110; why this is the case, however, has remained unclear.

To investigate this question, Sztangierska et al. used a variety of biochemical assays to test how purified human and yeast Hsp70, Hsp110 and other co-chaperones could bind aggregates and recover misfolded proteins. The role of each protein was examined at every stage of the disaggregation process – from the initial aggregate binding, through chaperone-driven changes in aggregate structure to the final protein folding.

The experiments revealed that Hsp110 helps draw Hsp70 to the aggregate surface, breaking down the protein 'clump' into smaller pieces which are more easily processed by other chaperones. The results also showed that the various co-chaperones compete for Hsp70 binding; too much of one might interfere with another, emphasizing the need for balance between chaperones for optimal disaggregation.

Overall, these results clarify the role of Hsp110 in the Hsp70 system and reveal several mechanistic details of the protein rescue process. Further experiments will be needed to fully understand these dynamics and identify how they may be relevant to conditions in which harmful protein aggregates are observed, such as Parkinson's or Alzheimer's disease.

whose J-domain induces ATP hydrolysis in Hsp70, resulting in conformational changes that stabilise the interaction with aggregates (*Rohland et al., 2022*). Next, Hsp100 interacts with the Hsp70-aggregate complex and the aggregate-trapped polypeptides are translocated through the Hsp100 hexamer. The disentangled and released polypeptides can fold back into their native conformation, alone or with further aid of chaperones (*Lum et al., 2004*; *Schaupp et al., 2007*; *Seyffer et al., 2012*; *Weibezahn et al., 2004*; *Zietkiewicz et al., 2004*).

Hsp110 co-chaperone boosts the Hsp70 activity by stimulating nucleotide exchange and substrate release (*Dragovic et al., 2006*; *Raviol et al., 2006*; *Shaner et al., 2005*). It belongs to the Hsp70 superfamily, with identical domain organisation to Hsp70 but distinct size and arrangement of the nucleotide-binding domain (NBD) and substrate-binding domain (SBD) (*Easton et al., 2000*; *Liu and Hendrickson, 2007*). In contrast to Hsp70, Hsp110 is unable to refold denatured proteins, yet it may bind to and prevent aggregation of certain misfolding substrates (*Garcia et al., 2017*; *Glover and Lindquist, 1998*; *Goeckeler et al., 2002*; *Oh et al., 1997*; *Oh et al., 1999*; *Polier et al., 2010*; *Xu et al., 2012*). In its well-established NEF function, the NBD of Hsp110 interacts with the ADP form of Hsp70, which mediates the release of the nucleotide. Subsequent ATP binding to the NBD of Hsp70 prompts the opening of the SBD and substrate dissociation, resetting Hsp70 for another round of the cycle (*Andréasson et al., 2008a*; *Laufen et al., 1999*; *Liberek et al., 1991*; *Mayer and Bukau, 2005*; *Mayer and Gierasch, 2019*).

In vivo studies on yeast chaperones demonstrated that the Hsp110 co-chaperone Sse1 plays a key role in Hsp70 recruitment to aggregates and is essential for protein disaggregation (*Kaimal et al., 2017*). Hsp110 is a major cytoplasmic NEF of Hsp70 and the deletion of its both paralogs, *SSE1* and *SSE2*, is lethal and the growth can only be supported, albeit with less efficacy, by overexpression of *FES1,* another cytoplasmic NEF from an unrelated armadillo family (*Abrams et al., 2014*; *Raviol et al., 2006*).

The importance of Hsp110 is also manifested in vitro in the low disaggregation activity of the yeast Hsp70 system without Hsp110 and the complete interdependence between their human orthologs (*Glover and Lindquist, 1998*; *Mattoo et al., 2013*; *Nillegoda et al., 2015*; *Rampelt et al., 2012*). Recent studies on the human system revealed that Hsp110 function in disaggregation of amyloid fibrils is not limited to its NEF activity but it may also affect the architecture of chaperone complexes, inducing Hsp70 clustering, although it is not clear whether similar effects occur during disaggregation

of non-fibrillar, stress-associated aggregates. The effect is not well understood, but it presumably increases entropic pulling of aggregated polypeptides, ultimately leading to their disentanglement by the Hsp70 system, efficient even without the Hsp100 disaggregase. Developing an Hsp70 system that is self-sufficient in disaggregation could have compensated for the loss of the Hsp100 disaggregase in a common metazoan ancestor (*Mattoo et al., 2013*; *Shorter, 2011*).

Another factor that potentiates the Hsp70 system is the diversity of JDP paralogs, assigned to cytoplasmic classes A and B, which differently regulate Hsp70 (*Lu and Cyr, 1998*; *Nillegoda et al., 2015*). The main distinction between class A and B JDPs is an auxiliary interaction site between CTD1 domain of class B JDPs and the C-terminal EEVD motif of Hsp70 (*Yu et al., 2015*). Class B JDPs are additionally regulated by an autoinhibitory mechanism, in which Hsp70 binding by the J-domain is restricted by a neighbouring helix. Upon binding to the C-terminal EEVD motif of Hsp70, the J-domain is released and it can interact with the NBD of Hsp70 (*Faust et al., 2020*; *Wentink et al., 2020*). We have recently shown that the yeast class A JDP Ydj1 and class B JDP Sis1 exhibit diverse mechanisms during Hsp70 binding to aggregated substrates. Ydj1, in accordance with the classical model of the Hsp70 ATPase cycle, binds misfolded polypeptides and loads Hsp70 onto aggregates. Unlike that, Sis1 only weakly binds protein substrates but due to the more complex interaction with Hsp70, it loads more Hsp70 molecules onto aggregates, which results in more efficient disaggregation (*Wyszkowski et al., 2021*).

Despite the vast knowledge on how Hsp110 serves as a regulator of the ATPase cycle of Hsp70 (*Andréasson et al., 2008a*; *Andréasson et al., 2008b*; *Dragovic et al., 2006*; *Raviol et al., 2006*; *Shaner et al., 2006*), little is known about the function of Hsp110 considering the mechanism of Hsp70 interaction with different JDP classes.

Here, we demonstrate that the interplay between Hsp110 and Hsp70 in disaggregation strictly relies on the class of a JDP and unravel the critical role of the B-class-specific interaction with the EEVD motif of Hsp70 for the Hsp110-dependent stimulation. Furthermore, we elucidate differential contribution of the NEF across different phases of protein recovery from aggregates. We employ biolayer interferometry (BLI) to investigate the Hsp110 impact on the formation of chaperone complexes at the aggregate surface and we assess changes in aggregates' properties associated with the abundant chaperone binding. Finally, we address a question of the competition between the NEF and JDP co-chaperones. Our findings shed new light on the mechanisms behind the potentiation and inhibition of the disaggregation activity of Hsp70 by its co-chaperones.

## Results

### Stimulation of the Hsp70 disaggregation activity by Hsp110 depends on the class of JDP

Our recent studies showed that during the recovery of aggregated proteins, the Hsp70 chaperone system exhibits different mechanisms of action with class A and class B JDP co-chaperones (*Wyszkowski et al., 2021*). To better understand the interplay within the Hsp70 chaperone network, we addressed how Hsp110 affects Hsp70 with members of different JDP classes. First, we tested the effect of Sse1, the most abundant yeast NEF, on the recovery of a model protein substrate, aggregated luciferase, by Hsp70 (Ssa1) and class A (Ydj1) or class B (Sis1) JDPs. Compared with Ssa1-Sis1, which exhibits delayed start of disaggregation characteristic for class B JDPs, the initial luciferase recovery was significantly faster, with higher overall output, in the presence of Sse1 (*Figure 1A*). In contrast, Sse1 slightly decreased the disaggregation efficacy in the case of Ssa1-Ydj1 (*Figure 1A*). With another aggregated substrate, GFP, Sse1 improved the disaggregation activity with either of the JDPs, albeit for Ssa1-Sis1, the stimulation was almost four times higher than with Ydj1 (*Figure 1— figure supplement 1A*).

Addition of the Hsp104 disaggregase to the system increased the total disaggregation efficacy and enhanced the positive effects of Sse1 in the case of either of JDPs and either of aggregated substrates (*Figure 1—figure supplement 1B and C*). The chaperone system comprising Hsp104-Ssa1-Sis1 with Sse1 yielded approximately three times higher luciferase and GFP recovery rates than without the NEF, while in the case of Hsp104-Ssa1-Ydj1, the simulation by Sse1 was much weaker, from marginal to twofold, depending on the protein substrate (*Figure 1—figure supplement 1B and C*). Thus, the positive effect of the Hsp110 NEF on protein recovery from aggregates is substantially more pronounced when the Hsp70 chaperone system comprises a class B JDP.

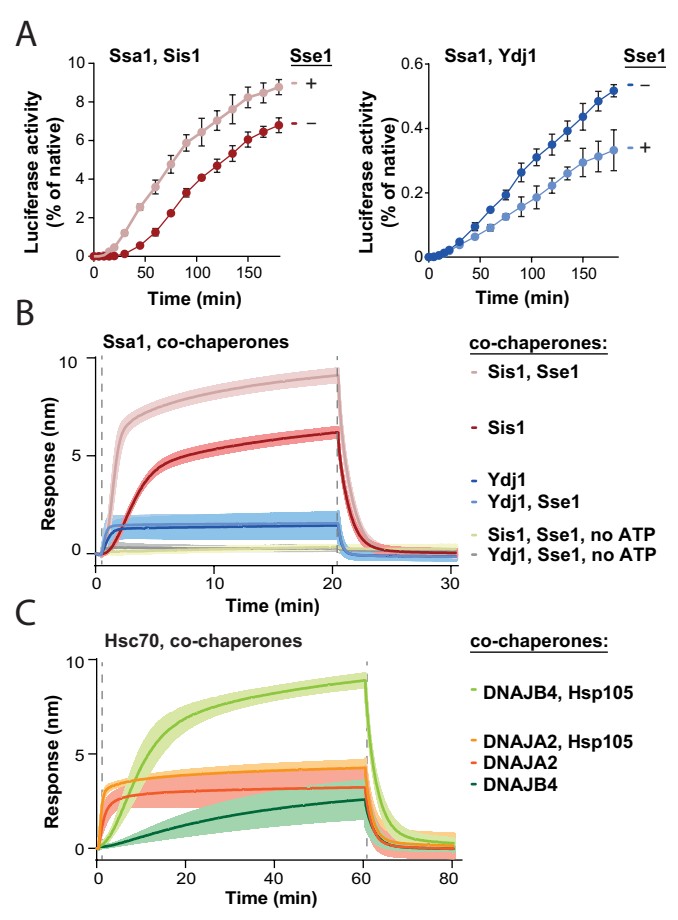

**Figure 1.** Impact of Hsp110 on protein disaggregation by Hsp70 system depends on class of J-domain protein (JDP). (**A**) Refolding of aggregated luciferase by Ssa1-Sis1 (1 µM Ssa1, 1 µM Sis1)±µM Sse1 (0.1 µM) (left) or Ssa1-Ydj1 (1 µM Ssa1, 1 µM Ydj1)±Sse1 (0.1 µM) (right). Error bars show SD from three independent repeats. Luciferase activity was measured at indicated time points and normalised to the native activity. (**B**) Sensor-bound luciferase aggregates incubated with Ssa1-Sis1 (1 µM Ssa1, 1 µM Sis1)±Sse1 (0.1 µM) or Ssa1-Ydj1 (1 µM Ssa1, 1 µM Ydj1)±Sse1 (0.1 µM), with or without ATP, as indicated. (**C**) Binding of Hsc70-DNAJB4 (3 µM Hsc70, 1 µM DNAJB4) or Hsc70-DNAJA2 (3 µM Hsc70, 1 µM DNAJA2) with or without Hsp105 (0.3 µM) to the heat-aggregated luciferase immobilised on the biolayer interferometry (BLI) biosensor. (**B, C**) The lines represent the average of three replicates, the shades designate SD, and the dashed lines indicate the start of the chaperone binding and dissociation steps.

The online version of this article includes the following source data and figure supplement(s) for figure 1:

**Source data 1.** Spreadsheet containing data for the graphs shown in *Figure 1A*.

**Source data 2.** Spreadsheet containing data for the graph shown in *Figure 1B*.

**Source data 3.** Spreadsheet containing data for the graph shown in *Figure 1C*.

**Figure supplement 1.** J-domain protein-specific impact of Hsp110 on protein disaggregation by Hsp70 system.

**Figure supplement 1—source data 1.** Spreadsheet containing data for the graphs shown in *Figure 1—figure supplement 1A*.

**Figure supplement 1—source data 2.** Spreadsheet containing data for the graphs shown in *Figure 1—figure supplement 1B*.

**Figure supplement 1—source data 3.** Spreadsheet containing data for the graphs shown in *Figure 1—figure supplement 1C*.

**Figure supplement 1—source data 4.** Spreadsheet containing data for the graph shown in *Figure 1—figure supplement 1D*.

**Figure supplement 1—source data 5.** Spreadsheet containing data for the graph shown in *Figure 1—figure*

*Figure 1 continued*

*supplement 1E* and another replicate of the experiment.

**Figure supplement 1—source data 6.** Spreadsheet containing data for the graph shown in *Figure 1—figure supplement 1F* and another replicate of the experiment.

**Figure supplement 1—source data 7.** Spreadsheet containing data for the graph shown in *Figure 1—figure supplement 1G* and another replicate of the experiment.

**Figure supplement 1—source data 8.** Spreadsheet containing data for the graphs shown in *Figure 1—figure supplement 1H*.

**Figure supplement 1—source data 9.** Spreadsheet containing data for the graph shown in *Figure 1—figure supplement 1I*.

**Figure supplement 1—source data 10.** Spreadsheet containing data for the graph shown in *Figure 1—figure supplement 1J* and another replicate of the experiment.

**Figure supplement 2.** Hsp110 determines level of Hsp70 binding to aggregates.

**Figure supplement 2—source data 1.** Spreadsheet containing data for the graphs shown in *Figure 1—figure supplement 2A*.

**Figure supplement 2—source data 2.** Spreadsheet containing data for the graph shown in *Figure 1—figure supplement 2B*.

**Figure supplement 3.** Effects of Fes1 on Hsp70 binding to protein aggregates and disaggregation.

**Figure supplement 3—source data 1.** Spreadsheet containing data for the graph shown in *Figure 1—figure supplement 3A*.

**Figure supplement 3—source data 2.** Spreadsheet containing data for the graph shown in *Figure 1—figure supplement 3B*.

**Figure supplement 3—source data 3.** Spreadsheet containing data for the graphs shown in *Figure 1—figure supplement 3C*.

---

Theoretically, the stimulation by Sse1 might occur at one or many stages of protein disaggregation – starting from the association of the Hsp70 system with an aggregate surface, through polypeptide disentanglement, to the final folding of the released substrate. Previous studies show that Ssa1-Ydj1 is much more effective than Ssa1-Sis1 in restoring the native structure of unfolded luciferase (*Lu and Cyr, 1998*), which could represent the final step of disaggregation – the polypeptide folding after translocation by Hsp104. To test how Sse1 contributes to the protein folding by the Hsp70 system, we measured the recovery of denatured, non-aggregated luciferase (*Imamoglu et al., 2020*). Curiously, neither of the systems, Ssa1-Sis1 nor Ssa1-Ydj1, was stimulated in luciferase folding by Sse1 (*Figure 1—figure supplement 1D*), suggesting that the NEF plays an important role at earlier stages of disaggregation.

We recently showed that JDP co-chaperones determine the association kinetics and the size of chaperone complexes formed at the aggregate surface, which contributes to the total effectiveness of disaggregation (*Wyszkowski et al., 2021*). To gain insight into the influence of the Hso110 on the assembly of the protein disaggregation complex, we analysed the binding of the Hsp70 system to heat-aggregated luciferase using BLI. The presence of Sse1 increased the rate of Ssa1-Sis1 association with an aggregate-covered biosensor approximately two times and resulted in a 50% thicker protein layer (*Figure 1B*). In contrast, Sse1 did not substantially influence the binding level of Ssa1-Ydj1 (*Figure 1B*). Similar effects were observed with heat-aggregated GFP or heat-aggregated yeast lysate proteins immobilised on the sensor, indicating that they are not substrate-specific (*Figure 1—figure supplement 1E and F*). The binding of chaperones to the aggregates required ATP (*Figure 1B*), which is consistent with the ATP dependence of Hsp70 (*Figure 1B*). Sse1 did not stimulate the binding of Ssa1 or JDPs alone, nor did it interact with the aggregate on its own (*Figure 1—figure supplement 1G*).

We also assessed the evolutionary conservation of the observed trends and applied BLI to investigate how Hsp105, a human Hsp110, affects aggregate binding by the Hsp70 system. Similarly as the yeast system, Hsc70-DNAJB4 (orthologous to Ssa1-Sis1) exhibited delayed binding and the addition of Hsp105 resulted in much faster association and a sixfold increase in the binding level (*Figure 1C*). In the case of Hsc70-DNAJA2 (orthologous to Ssa1-Ydj1), the binding also increased upon Hsp105 addition, however, the stimulation was three times less pronounced than with DNAJB4 (*Figure 1C*). In

contrast to the yeast proteins, the stimulation occurred with both JDP classes, which is in line with the stronger dependence of the human Hsp70 system on the NEF in protein disaggregation (*Figure 1—figure supplement 1H*).

The BLI results suggest that the Hsp110 NEF greatly improves Hsp70 binding to aggregates, especially with class B JDPs. What could be the basis for this specificity? A major discrimination factor between the two JDP classes is the stable interaction between class B JDPs and the EEVD motif of Hsp70 (*Yu et al., 2015*). Perturbation of this interaction restricts the J-domain-dependent Hsp70 activation, which can be restored through the E50A mutation in Sis1 (*Yu et al., 2015*) or the mutation in DNAJB4 Helix 5 (*Faust et al., 2020*). Previous studies showed that Ssa1$^{\Delta EEVD}$-Sis1$^{E50A}$ resembles Ssa1-Ydj1 in aggregate binding and disassembly (*Wyszkowski et al., 2021*), therefore we asked what effect this disruption has on the stimulation by Sse1. We observed that the Sis1 E50A-Ssa1 ΔEEVD system is strongly inhibited by Sse1 in protein disaggregation (*Figure 1—figure supplement 1I*) and the level of aggregate binding is also negatively affected (*Figure 1—figure supplement 1J*). This suggests that the Hsp70-JDP stimulation by Hsp110 is functionally linked to the Sis1-Ssa1 EEVD interaction.

We next asked whether Hsp110 contributes to the extra chaperone layer at the aggregate surface in a way that it comprises more Hsp70 molecules. To evaluate the amount of bound Hsp70, we carried out an analogous BLI experiment with fluorescently labelled Ssa1 (Ssa1*A488). After protein dissociation to the basal level, the total dissociated Ssa1*A488 was quantified based on fluorescence. In the presence of Sis1 and Sse1, the amount of Ssa1*A488 was nearly two times higher than in the presence of Sis1 only, whereas Sse1 did not influence the amount Ssa1*A488 bound to the biosensor when Ydj1 was applied (*Figure 1—figure supplement 2A*). The increase in the fluorescence signal corresponded to that of BLI (*Figure 1—figure supplement 2A*), suggesting a major contribution of Ssa1 to the thickness of the chaperone complex forming on the aggregate surface.

Since the presence of Sse1 substantially increased the association of Ssa1 with aggregates, we asked if the problem limiting Hsp70 binding that is overcome by Hsp110 is the insufficient availability of Hsp70 molecules targeting the aggregate. To address that, we tested whether a similar effect to the Sse1-induced stimulation of Ssa1 binding can be obtained by increasing Ssa1 concentration itself. Sse1 stimulated Ssa1 binding much above the level that could be achieved for the saturating concentration of Ssa1 in the absence of Sse1 (*Figure 1—figure supplement 2B*). This indicates that the mechanism of stimulation by Hsp110 is more complex than an enrichment of the pool of Hsp70 molecules capable of substrate binding and might involve generating more Hsp70-binding sites.

To explore whether the observed effects are unique to Hsp110, we examined another cytosolic NEF that belongs to the Armadillo type family, Fes1. It is structurally distinct from Hsp110, with a C-terminal armadillo domain that triggers nucleotide exchange in Hsp70 through a different mechanism (*Gowda et al., 2018*). Fes1 has weaker affinity for Hsp70 and lower nucleotide exchange activity than Sse1 (*Dragovic et al., 2006*). Unlike Sse1, Fes1 does not directly bind protein substrates, but its N-terminal RD domain is involved in substrate release from Hsp70 (*Gowda et al., 2018*). When we carried out luciferase disaggregation by Ssa1-Sis1, the level of stimulation achieved by Sse1 required 10 times more Fes1 (*Figure 1—figure supplement 3A*). Similarly, 1 μM Fes1 stimulated Ssa1 binding to luciferase aggregates on the BLI sensor to the level achieved with 0.1 μM Sse1 (*Figure 1—figure supplement 3B and C*, *Figure 1—figure supplement 2A*). This result suggests that the substrate-binding activity specific to Hsp110 is not necessary to increase the Sis1-Ssa1 binding to aggregates and that the effective NEF concentration is negatively correlated with its affinity for Hsp70.

## Sse1 leads to Hsp70-dependent reduction of aggregate size

Recently, we have reported that larger Hsp70-JDP assemblies at the aggregate surface, dependent on Sis1 and its interaction with EEVD, can modify aggregates into misfolded protein species that are more amenable to disaggregation (*Wyszkowski et al., 2021*). To assess whether Hsp110 further stimulates such aggregate-remodelling activity, and this way contributes to the more efficient protein recovery, we used a variant of Hsp104 with abrogated interaction with Hsp70, D484K F508A (Hsp104$^{mut}$). Hsp104$^{mut}$ does not require Hsp70 for allosteric activation and aggregate binding and it can serve as an indicator of Hsp70-dependent aggregate modification by facilitating final reactivation of its products (*Chamera et al., 2019*). When heat-aggregated GFP was initially incubated with Ssa1-Sis1-Sse1, which yielded very low protein recovery, we added Hsp104$^{mut}$ and observed much faster

and more effective GFP reactivation than when the substrate was first incubated with Ssa1-Sis1 only (*Figure 2A*). In an analogous experiment with Ydj1, we also observed stimulation by Sse1, yet not as strong as in the presence of Sis1 (*Figure 2A*).

Aggregate remodelling by Hsp70-JDP, improved by Hsp110, might induce changes limited to the aggregate surface, such as partial polypeptide disentanglement that uncovers additional chaperone-binding sites, or also lead to global rearrangements, changing aggregate size and total exposed surface area, e.g., through partial aggregate dissolution and fragmentation. To shed light on this, we visualised the aggregates of luciferase C-terminally fused with GFP using fluorescence microscopy. After heat denaturation, approximately 80% of luciferase-GFP aggregates had the size of 2 μm or more (*Figure 2B*). Incubation with Ssa1-Sis1 for 1 hr decreased the fraction of aggregates larger than 2 μm to 25.8%, whereas the addition of Sse1 reduced it to 12.5% (*Figure 2B*). At the same time, the activity remained at the level of 1–2% of the native luciferase-GFP and only the presence of Hsp104-Hsp70 led to its substantial recovery (*Figure 2—figure supplement 1A*). The reduced size of luciferase-GFP aggregates by Hsp70 depended on ATP and the aggregate size was only moderately affected by Ssa1-Ydj1, irrespective of the presence of Sse1 (*Figure 2B*).

We also measured the size of aggregates using dynamic light scattering (DLS). Heat-denatured luciferase formed aggregates with hydrodynamic diameter of approximately 2000 nm (*Figure 2—figure supplement 1B*), while the DLS signal of approximately 10 nm corresponded to the native luciferase and chaperones, consistent with their theoretical size (*Figure 2—figure supplement 1B*). Incubation with Ssa1-Sis1 for 1 hr slightly reduced the 2000 nm peak and produced small-size aggregates of approximately 30 nm (*Figure 2—figure supplement 1B*). The presence of Sse1 increased the amount of 30 nm aggregates more than twofold and significantly reduced the signal at 2000 nm (*Figure 2—figure supplement 1B*). The change in aggregate size was not observed unless ATP was present (*Figure 2—figure supplement 1B*). In agreement with the microscopy data, Ssa1-Ydj1 with or without Sse1 did not have any considerable effect on the size of the aggregates (*Figure 2—figure supplement 1B*).

Taken together, Sse1 boosts the aggregate-remodelling activity of Ssa1, specifically with class B JDP Sis1, causing major reduction in aggregate size.

## Hormetic effects of Sse1 in Hsp70 disaggregation activity

Contrary to the strong stimulation by Hsp110 of the Hsp70 system comprising class B JDP, its impact on Hsp70 with class A JDP strongly varied with experimental setup (*Figure 1A and B*, *Figure 1—figure supplement 1A–C*). Knowing that the effects of Sse1 on the disaggregation by Ssa1-Ydj1 are concentration-dependent, with stimulation at sub-stoichiometric amount of Sse1 (*Dragovic et al., 2006*), we asked, to what degree the optimum Hsp110 level depends on the class of a JDP. When we titrated the Hsp70 system comprising either Sis1 or Ydj1, we observed an inhibition of luciferase disaggregation at increased Sse1 concentrations, yet the system with Ydj1 was much more sensitive to Sse1 (IC50=0.1 μM), while Ssa1-Sis1 was still stimulated at 0.3 μM Sse1 (molar ratio Ssa1:Sse1 1:0.3) (*Figure 3A*). When we included the Hsp104 disaggregase, both Hsp104-Ssa1-Ydj1 and Hsp104-Ssa1-Sis1 systems were stimulated at low and inhibited at high Sse1 levels, with the highest yield at 0.05 μM and 0.2 μM of Sse1, respectively (*Figure 3—figure supplement 1A*). A similar biphasic effect was observed for human orthologs, with Hsc70-DNAJB4 tolerating higher Hsp105 concentration than Hsc70-DNAJA2 (*Figure 3B*).

In contrast to protein disaggregation, the reactivation of denatured, non-aggregated luciferase by Ssa1 with either Sis1 or Ydj1 was only negatively affected across a range of Sse1 concentrations (*Figure 3C*), suggesting that the positive contribution of Hsp110 takes place before polypeptides get released from an aggregate. The negative effect of Sse1 on protein folding could potentially mask a positive effect of saturating Sse1 on aggregate modification, e.g., through entropic pulling. We thus analysed the size of aggregates after their incubation with Ssa1-Sis1 and 1 μM Sse1, however it was unaffected (*Figure 3—figure supplement 1B and C*), suggesting that higher Sse1 levels also limit the aggregate-remodelling activity of Hsp70. Consistently, when we measured Hsp70 association with luciferase aggregated on the BLI sensor across Sse1 concentrations, we observed the biphasic effect for Ssa1-Ydj1 and Ssa1-Sis1, with the maximum binding at 0.05 μM and 0.2 μM of Sse1, respectively (*Figure 3D*). Thus, Hsp110 promotes Hsp70 assembly at aggregates and their modification only at sub-stoichiometric concentrations, with peak performance at higher levels of Sse1 in the presence of class B than class A JDPs.

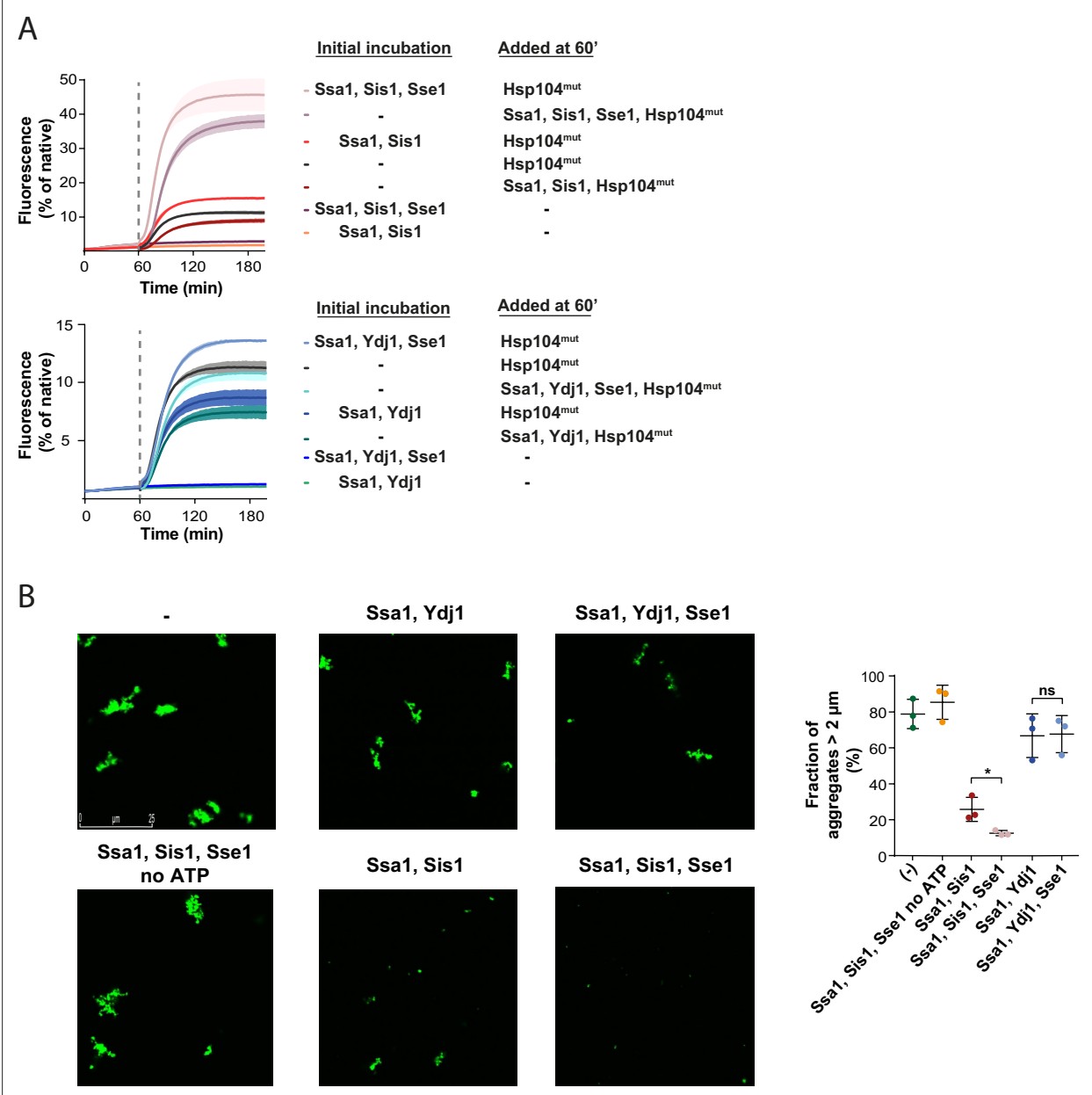

**Figure 2.** Sse1 promotes modification of aggregates by the Hsp70 system. (**A**) Initial incubation of heat-aggregated GFP aggregates with the Hsp70 system, followed by the addition of the Hsp104 D484K F508A variant (0.15 µM). Recovery was initiated by the addition of the mix of indicated chaperones: 1 µM Ssa1, 1 µM Sis1, 1 µM Ydj1, or 0.1 µM Sse1. Dashed lines indicate the beginning of the incubation with the Hsp104 variant. Curves show average values and shades indicate SD from three replicates. (**B**) Fluorescence microscopy images of luc-GFP monitored upon addition of Ssa1-Sis1±Sse1 or Ssa1-Ydj1±Sse1. Chaperones were used at 1 µM concentration, except for 0.1 µM Sse1. Left panels show controls of the luciferase-GFP aggregates alone and upon the addition of the Hsp70 system without ATP. Quantification of the fraction of aggregates >2 µm is from three independent replicates. Two-tailed t test was performed: *$p<0.05$, ns: not significant.

The online version of this article includes the following source data and figure supplement(s) for figure 2:

**Source data 1.** Spreadsheet containing data for the graph shown in *Figure 2A*.

**Source data 2.** Spreadsheet containing data for the graph shown in *Figure 2B*.

**Source data 3.** Uncropped microscopy images presented in *Figure 2B* and replicates for the calculations in *Figure 2—source data 2*.

**Figure supplement 1.** Hsp110 impact on aggregate modification by Hsp70 system.

**Figure supplement 1—source data 1.** Spreadsheet containing data for the graph shown in *Figure 2—figure supplement 1A*.

**Figure supplement 1—source data 2.** Spreadsheet containing data for the graphs shown in *Figure 2—figure supplement 1B*.

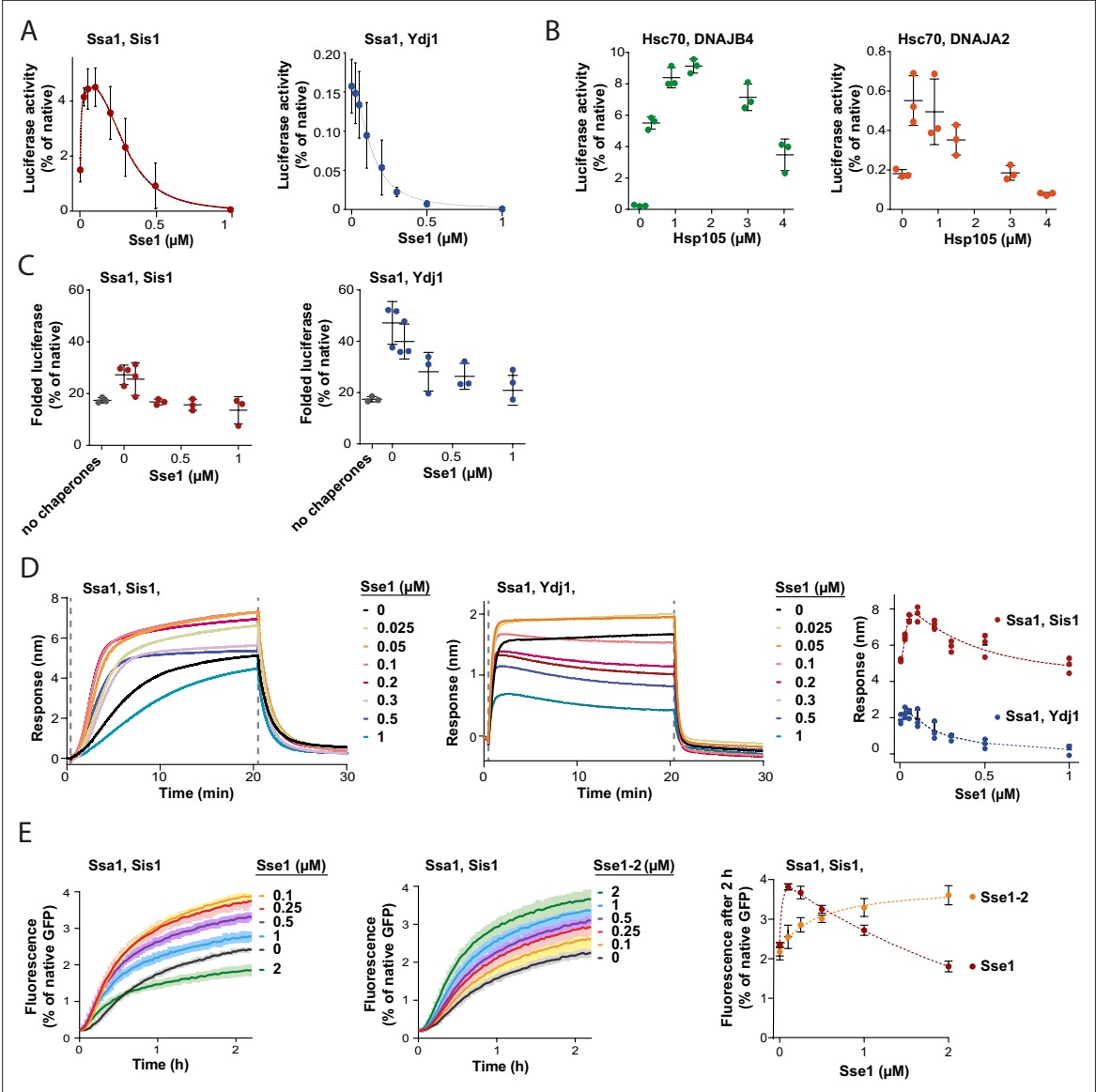

**Figure 3.** Susceptibility of Hsp70 to Hsp110 depends on J-domain protein (JDP) class and phase of disaggregation. (**A**) Titration of Sse1 in the refolding of aggregated luciferase by Ssa1-Sis1 (1 µM Ssa1, 1 µM Sis1) (red) or Ssa1-Ydj1 (1 µM Ssa1, 1 µM Ydj1) (blue). Activity of luciferase was measured after 1 hr and normalised to that of the native protein. (**B**) Incubation of Hsc70-DNAJB4 (3 µM Hsc70, 1 µM DNAJB4) (green) or Hsc70-DNAJA2 (3 µM Hsc70, 1 µM DNAJA2) (orange) with luciferase aggregates at increasing concentrations of Hsp105. Luciferase activity was measured after 4 hr and normalised to the activity of the native protein. (**C**) Folding of non-aggregated luciferase diluted from 5 M GuHCl (grey), spontaneous or mediated by the Hsp70 system comprising Ssa1-Sis1 (1 µM Ssa1, 1 µM Sis1) (red) or Ydj1-Ssa1 (1 µM Ssa1, 1 µM Ydj1) (blue) with increasing concentrations of Sse1. Activity of luciferase was measured after 2 hr and normalised to the native protein. (**D**) Binding of Ssa1-Sis1 (1 µM Ssa1, 1 µM Sis1) or Ssa1-Ydj1 Ssa1 (1 µM Ssa1, 1 µM Ydj1) in the presence of Sse1 at the indicated concentrations to the sensor covered with luciferase aggregates. Right panel shows a plot of the binding signal prior to the dissociation step. (**E**) Renaturation of heat-aggregated GFP by Ssa1-Sis1 (1 µM Ssa1, 1 µM Sis1) in the presence of Sse1 or Sse1-2 at the indicated concentrations. Right panel shows the plot of the recovered GFP activity after 2 hr of incubation with the Hsp70 system in the presence of Sse1 (orange) or Sse1-2 (red). (**D, E**) Dashed lines show the fitting of the *[Agonist] vs response* model to the data from the stimulation and inhibition phases separately using the GraphPad Prism software. (**A–E**) Error bars and shades indicate SD from three repeats.

The online version of this article includes the following source data and figure supplement(s) for figure 3:

**Source data 1.** Spreadsheet containing data and model fitting parameters for the graphs shown in *Figure 3A*.

**Source data 2.** Spreadsheet containing data for the graphs shown in *Figure 3B*.

**Source data 3.** Spreadsheet containing data for the graphs shown in *Figure 3C*.

**Source data 4.** Spreadsheet containing data for the graphs shown in the left and middle panels of *Figure 3D*.

*Figure 3 continued on next page*

*Figure 3 continued*

**Source data 5.** Spreadsheet containing data for the graph shown in the right panel of *Figure 3D*.

**Source data 6.** Spreadsheet containing data for the graphs shown in the left and middle panels of *Figure 3E*.

**Source data 7.** Spreadsheet containing data and model fitting parameters for the graph shown in the right panel of *Figure 3E*.

**Figure supplement 1.** Concentration-dependent effects of Hsp110 on aggregate binding and disaggregtion by Hsp70.

**Figure supplement 1—source data 1.** Spreadsheet containing data for the graphs shown in *Figure 3—figure supplement 1A*.

**Figure supplement 1—source data 2.** Spreadsheet containing data for the graph shown in *Figure 3—figure supplement 1B*.

**Figure supplement 1—source data 3.** Uncropped microscopy images presented in *Figure 3—figure supplement 1B* and replicates for the calculations in *Figure 3—figure supplement 1—source data 2*.

**Figure supplement 1—source data 4.** Spreadsheet containing data for the graph shown in *Figure 3C*.

**Figure supplement 1—source data 5.** Spreadsheet containing data for the graphs shown in *Figure 3D*.

**Figure supplement 1—source data 6.** Spreadsheet containing data for the graph shown in *Figure 3E*.

**Figure supplement 1—source data 7.** Spreadsheet containing data for the graph shown in *Figure 3F*.

To establish whether the inhibition that occurs at higher Sse1 concentrations is directly associated with the interaction between Sse1 and Ssa1, or rather between Sse1 and a substrate, we used the previously characterised Sse1-2 variant (N572Y E575A), with disrupted interaction with the A300 residue of Ssa1, which reduces Hsp70 binding to 20% and nucleotide exchange to 5% of that of the wild type (WT) (*Polier et al., 2008*). The variant has been reported to partially compensate for the *SSE1/SSE2* double deletion (*Polier et al., 2008*). Consistently with the weak Ssa1 binding affinity of Sse1-2 (*Figure 3—figure supplement 1D*), Ssa1-Sis1 required 20 times higher concentration of the Sse1-2 variant than Sse1 WT to reach the same GFP disaggregation activity and no inhibition was observed up to 2 µM of Sse1-2 (*Figure 3E*). Also, the Sse1-2 variant had a much weaker impact on luciferase disaggregation (*Figure 3—figure supplement 1E*). Nonetheless, 1 µM Sse1-2 was sufficient to obtain a similar effect as 0.1 µM WT Sse1 in stimulating Ssa1-Sis1 binding to the aggregated luciferase in the BLI assay (*Figure 3—figure supplement 1F*).

This suggests that it is the high-affinity interaction between Hsp70 and the NEF that enables strong stimulation already at very low Hsp110 levels. On the other hand, such strong interaction is associated with the inhibition of the disaggregation activity when Hsp110 concentration exceeds sub-stoichiometric proportion to Hsp70.

## Hsp110 limits JDP interaction with Hsp70

Since all NEFs induce polypeptide release from Hsp70, excessive dissociation from the substrate seems the most apparent explanation of Hsp70 inhibition by Hsp110. However, the strong dependence of the impact of Hsp110 on the class of JDP prompted us to search for other potential mechanisms behind the low Hsp70 chaperone activity under high Hsp110 concentration. Knowing that the sensitivity to Sse1 (*Figure 3A and D*) is negatively correlated with the strength of the Hsp70-JDP interaction, higher in the case class B JDP due to its binding of the EEVD motif of Hsp70 (*Wyszkowski et al., 2021*), we explored the possibility that Hsp110 restricts the formation of the Hsp70-JDP complex.

To test this, we immobilised Sis1 on the BLI sensor and monitored Ssa1 binding across a range of Sse1 concentrations. The incubation with 0.1 µM Sse1 was almost inert, however, at 1:1 Ssa1:Sse1 ratio, the binding greatly diminished (*Figure 4A*). The reduced affinity of Sse1-2 for Ssa1 resulted in its milder negative effect on Ssa1 binding to Sis1 (*Figure 4—figure supplement 1A*), in agreement with its impact on disaggregation (*Figure 3E*). The degree of inhibition by the two Sse1 variants (*Figure 4—figure supplement 1B*) correlated with their capacity to exchange nucleotides (*Dragovic et al., 2006*; *Polier et al., 2008*).

In an analogous way, we analysed the interaction between the human Hsc70 and sensor-bound DNAJB4. With an increasing Hsp105 concentration, the binding signal declined to reach 85% at 1:1 Hsc70:Hsp105 ratio (*Figure 4—figure supplement 1C*), indicating that the negative Hsp110 impact on the Hsp70-JDP interaction exhibits a similar pattern across Fungi and Metazoa.

Since Sse1 inhibits Ssa1 binding to Sis1, we asked whether the disaggregation rate limitation imposed by Sse1 depends on the level of Sis1. To verify this, we incubated GFP aggregates with the

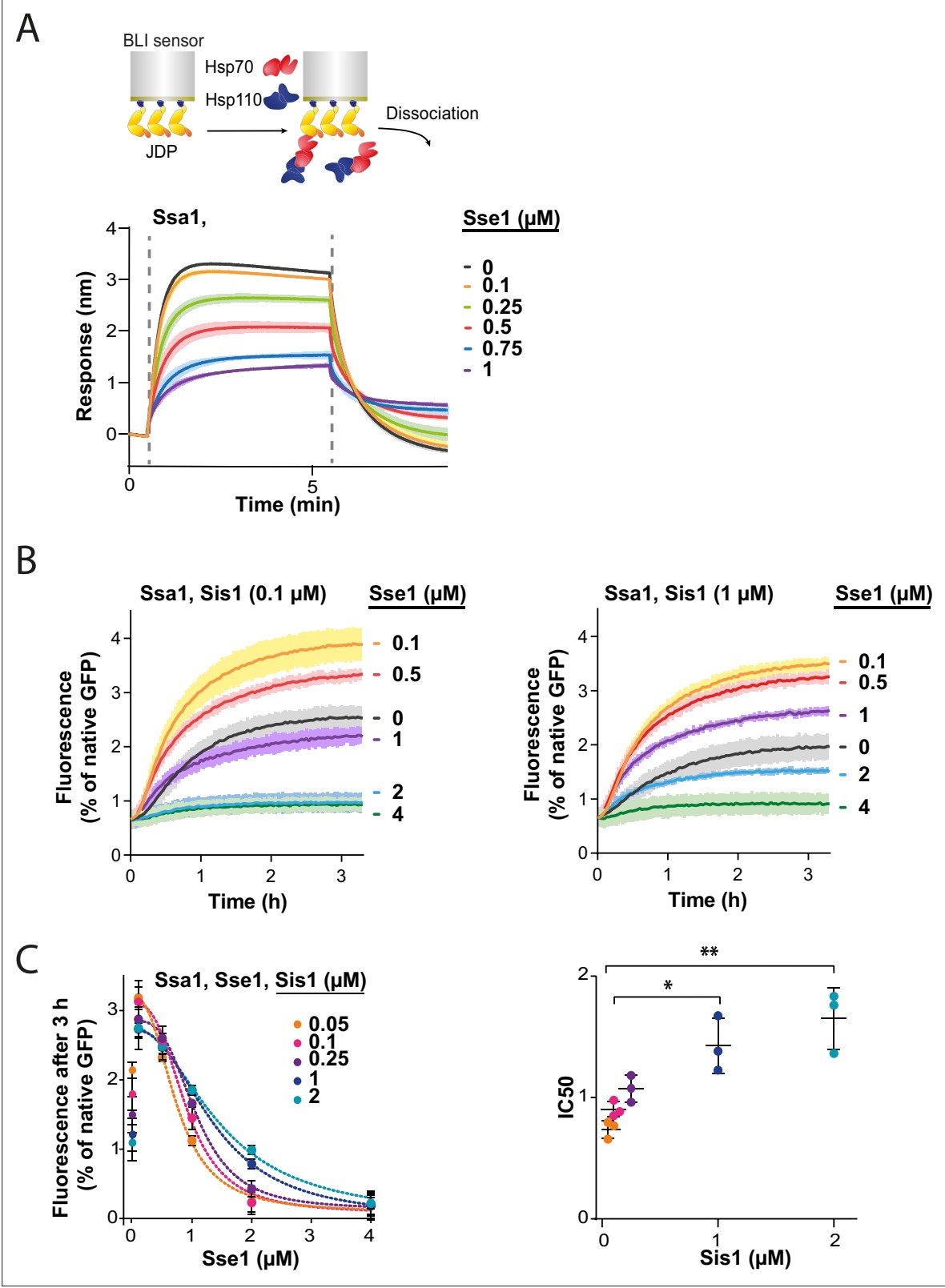

**Figure 4.** Hsp110 and class B J-domain protein (JDP) show apparent competition for Hsp70. (**A**) Upper panel shows the scheme of the biolayer interferometry (BLI) experiment. Binding of Ssa1 (1 µM) in the presence of increasing concentrations of Sse1 to Sis1 immobilised on the BLI sensor through the His₆-SUMO tag. Dashed lines indicate the moment of addition of chaperones to the sensor-bound Sis1 and the dissociation step. (**B**) Renaturation of heat-aggregated GFP by Ssa1-Sis1 (1 µM Ssa1, 0.1 µM, or 1 µM Sis1) at increasing concentrations of Sse1. (**C**) Plot of GFP activity

*Figure 4 continued on next page*

*Figure 4 continued*

after 3 hr recovered from aggregates by Ssa1 at different concentrations of Sis1 and Sse1 (left). IC50 was determined by fitting the [*Inhibitor] versus response* model to the data from three experiments using the GraphPad Prism software (dashed lines). Two-tailed t test: *p<0.05, **p<0.01. Lines are the average of three replicates and the error bars and shades designate standard deviation.

The online version of this article includes the following source data and figure supplement(s) for figure 4:

**Source data 1.** Spreadsheet containing data for the graph shown in *Figure 4A*.

**Source data 2.** Spreadsheet containing data for the graphs shown in *Figure 4B*.

**Source data 3.** Spreadsheet containing data and model fitting parameters for the graphs shown in *Figure 4C*.

**Figure supplement 1.** Competition between Hsp110 and JDP co-chaperones.

**Figure supplement 1—source data 1.** Spreadsheet containing data for the graph shown in *Figure 4—figure supplement 1A*.

**Figure supplement 1—source data 2.** Spreadsheet containing data for the graph shown in *Figure 4—figure supplement 1B*.

**Figure supplement 1—source data 3.** Spreadsheet containing data for the graph shown in *Figure 4—figure supplement 1C*.

**Figure supplement 1—source data 4.** Spreadsheet containing data for the graphs shown in *Figure 4—figure supplement 1D*.

**Figure supplement 1—source data 5.** Spreadsheet containing data for the graph and model fitting parameters shown in *Figure 4—figure supplement 1E*.

Hsp70 system comprising various Sse1 and Sis1 concentrations (*Figure 4B and C*, *Figure 4—figure supplement 1D*). The susceptibility of the Hsp70 system to elevated Sse1 concentration significantly decreased with increasing Sis1 concentration and at 2 μM Sis1, the stimulation of the initial disaggregation rate was still observed at 2 μM Sse1 (*Figure 4—figure supplement 1D*). Interestingly, in the absence of Sse1, the disaggregation rate dropped with increasing Sis1 concentrations and the trend was reversed at higher Sse1 levels, with an inflection point at approximately 0.5 μM of Sse1 (*Figure 4—figure supplement 1E*).

Thus, Sse1 and Sis1 exhibit an apparent competition for Ssa1 binding. These results indicate that the balance between the JDPs and NEF co-chaperones is critical for the performance of the Hsp70 chaperone system in protein disaggregation.

## Discussion

Our results uncover a comprehensive picture of the role of Hsp110 co-chaperones in aggregate processing by the Hsp70 system. Potentiation of the Hsp70 activity by the NEF strongly depends on the class of the JDP and the preference for class B, conserved in yeast and human orthologs, relies on Hsp70 binding through the EEVD motif. Furthermore, we demonstrate that the Hsp110-dependent stimulation of disaggregation is limited to its initial stages: chaperone recruitment to aggregates and their disassembly, but not final protein folding (*Figure 5*). The initial Hsp110-dependent loading of more Hsp70 onto aggregates correlates with their remodelling into smaller aggregated species, which improves their recognition by the Hsp104 disaggregase. We also gained insight into the biphasic impact of Hsp110: with increasing Sse1 level, the stimulation is overshadowed by inhibition and the contribution of each trend depends on the phase of disaggregation and the composition of the disaggregating system, with a crucial role of the NEF's affinity for Hsp70. Finally, we show that the disaggregation inhibition by Hsp110 involves disruption of the Hsp70-JDP interaction, suggesting competition between JDP and Hsp110 co-chaperones.

The balance between the partners within the Hsp70 system is key for efficient disaggregation, as an individual co-chaperone with a critical function at one step may inhibit another. To investigate this differential contribution, we used biochemical assays reflecting individual stages of protein disaggregation. The results of the BLI experiments show that Hsp110 with class B JDP (JDP[B]) greatly increases Hsp70 aggregate binding (*Figure 1B and C*, *Figure 1—figure supplements 1E, F and 2A*), which correlates with the final reactivation yield (*Figure 1A*, *Figure 1—figure supplement 1H*), implying that the initiation phase determines the overall disaggregation efficiency. On the other hand, the final folding of a soluble substrate by Ssa1 relies much more on the class A JDP (JDP[A]) Ydj1 than on Sis1 (*Figure 1—figure supplement 1D*; *Lu and Cyr, 1998*), and not on Sse1 altogether (*Figure 1—figure supplement 1D*), which is in line with previous reports that its human ortholog, Hsp105 inhibits luciferase folding (*Rauch and Gestwicki, 2014*).

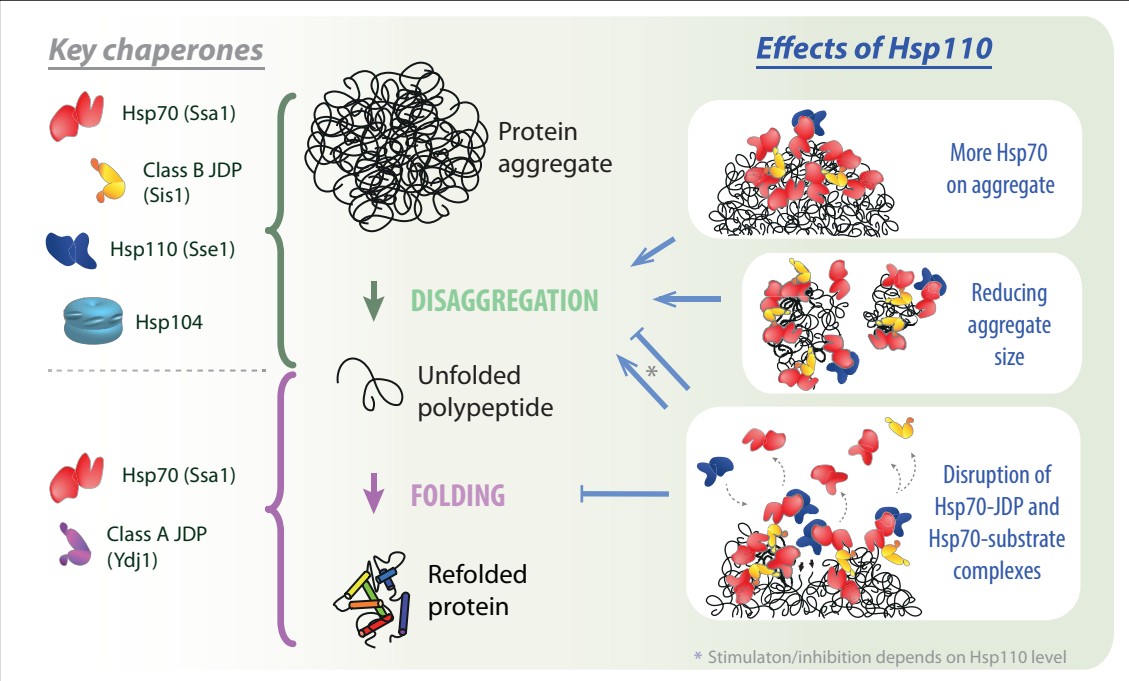

**Figure 5.** Impact of Hsp110 on Hsp70-dependent disaggregation. Hsp110 stimulates aggregate disassembly by Hsp70 with class B J-domain proteins (JDPs) and Hsp104 (green arrow), contrary to the stage of final folding of the solubilised polypeptide (light purple arrow), at which Hsp70 with class A JDP are most effective. Hsp110 improves the disaggregation by increasing Hsp70 binding to the aggregate (upper right panel) and mediates aggregate remodelling into smaller assemblies (middle right panel). By facilitating partial Hsp70 dissociation from the substrate and/or JDP, at the optimal, sub-stoichiometric concentration, Hsp110 might gradually uncover new Hsp70-binding sites (lower right panel, small black arrows), potentially leading to the more abundant and effective Hsp70 recruitment to the aggregate. At higher Hsp110 levels, the destabilisation of Hsp70 interactions with protein substrates and JDPs leads to the inhibition across all stages of protein disaggregation (lower right panel).

The higher degree of stimulation with class B JDP has been previously observed, albeit without broader insight, for Hsp110 and other NEF BAG (*Rampelt et al., 2012*; *Rauch and Gestwicki, 2014*). We show that the distinguishing feature fundamental for this specificity is the auxiliary interaction between the CTD domain of Sis1 and the EEVD motif of Hsp70. The Sis1 E50A variant, featuring dere-pressed J-domain (*Yu et al., 2015*), together with Ssa1 ΔEEVD exhibited a similar aggregate binding pattern with or without Sse1 to the system with Ydj1 (*Figure 1B*, *Figure 1—figure supplement 1E, F, and J*). However, while Ydj1 can bind misfolded proteins and prevent aggregation, Sis1 cannot (*Lu and Cyr, 1998*). When Sis1 is deprived of Ssa1 binding through EEVD, which has a major contribution to the stability of their interaction (*Wyszkowski et al., 2021*), the effect of Sse1 on disaggregation is detrimental (*Figure 1—figure supplement 1I*). This underlines the importance of the Hsp70-JDP complex stability for the stimulation of substrate binding and disaggregation by Sse1.

In our previous work, we reported sigmoidal aggregate binding kinetics characteristic for the Ssa1-Sis1 system, suggesting that the binding gradually generates more chaperone binding sites (*Wyszkowski et al., 2021*). Now, we show that this apparent cooperativity is conserved in yeast and human, implying mechanistic similarities between the systems (*Figure 1B and C*). While the s-shaped Ssa1-Sis1 binding to the aggregate results in the association of more Ssa1 than with Ydj1 (*Figure 1—figure supplement 2A*), Hsp110 shortens the initial lag phase and boosts the binding efficacy, loading a thicker layer of Hsp70 molecules onto aggregates (*Figure 1B and C*, *Figure 1—figure supplement 2A*), similarly as has been shown for amyloid fibrils (*Beton et al., 2022*). This corresponds with Hsc70 clustering on amyloid in the presence of Hsp110, reported by *Wentink et al., 2020*. Clustered, densely packed Hsp70 has been proposed to generate entropic pulling effect that leads to fibril fragmentation (*Goloubinoff and De Los Rios, 2007*; *Sousa and Lafer, 2019*; *Wentink et al., 2020*). Likewise, we observed remodelling of amorphous aggregates into smaller aggregated species, which resembles amyloid disassembly in terms of the dependence on the NEF activity of Hsp110, the class B JDP and its interaction with the Hsp70 EEVD motif (*Figure 2B*, *Figure 2—figure supplement 1B*;

*Beton et al., 2022*). We propose that the local extraction of polypeptides by Hsp70-JDP$^B$-Hsp110, which penetrates the surface and takes apart amorphous aggregates, although not enough for full protein recovery, could successfully generate more manageable substrates for chaperones, as we demonstrated by applying the derepressed Hsp104 variant (*Figures 2A and 5*). This Hsp70 activity, albeit yielding very low protein reactivation (*Figure 2—figure supplement 1A*), might be relevant for the fragmentation of a thinner, linear amyloid structure.

Such clustering-augmented aggregate remodelling and fragmentation, possibly through expanding the total effective aggregate surface, leads to the much-improved protein reactivation by Hsp104 (*Figure 1—figure supplement 1B and C*, *Figure 2—figure supplement 1A*). This would explain the previous findings that although Sse1 is not necessary for Hsp104 recruitment to aggregates, it is essential for Hsp104-dependent aggregate clearance in the cell (*Kaimal et al., 2017*). Apparently, despite the more heterogenic and dynamic nature of cellular aggregates than of those generated in vitro, their disaggregation by Hsp104 largely relies on the Hsp110-boosted processing by the Hsp70 system.

Above a certain Hsp110 level, however, the disaggregation is inhibited. The susceptibility to the NEF depends on the phase of the process and the composition of the Hsp70 system (*Figure 3A–D*, *Figure 3—figure supplement 1A*), which may explain the discrepancies between different studies of Hsp110's impact on disaggregation (*Dragovic et al., 2006*; *Polier et al., 2008*; *Raviol et al., 2006*; *Shorter, 2011*) As well established, the most effective is sub-stoichiometric Hsp110 proportion to Hsp70 (*Gao et al., 2015*; *Polier et al., 2008*; *Wentink et al., 2020*) which corresponds with the physiological conditions, where the ratio between all cytoplasmic Hsp70 and Hsp110 paralogs involved in disaggregation ranges from approximately 10:1.7 in yeast (*Brownridge et al., 2013*) to 10:2.4 in humans (*Finka and Goloubinoff, 2013*). According to our data, the optimum, at which the stimulation and inhibition curves intersect, is shifted to higher levels: (a) during the initiation versus final protein folding (*Figure 3A and C*), (b) with class B JDP, in comparison with class A (*Figure 3A, B, and D*), (c) with Hsp104, comparing with the system without the disaggregase (*Figure 3A*, *Figure 3—figure supplement 1A*), and (d) for the human Hsp70 system, comparing with the yeast chaperones (*Figure 3A and B*). The latter trend might result from the stronger dependence of the human Hsp70 chaperone on the nucleotide exchange activity, possibly associated with the lower basal rate of nucleotide dissociation than that reported for Ssa1 (*Dragovic et al., 2006*). Possibly, due to these problems with ADP release, the positive impact of Hsp110 on Hsc70 is manifested both in the presence of class A and B JDPs and the inhibition occurs at much higher Hsp110 levels. Accordingly, it is tempting to speculate that the human system, lacking Hsp104, has evolved towards better tolerance for Hsp110 to boost the disaggregation activity of the Hsp70 system alone, e.g., to counteract amyloid toxicity.

Excessive dissociation of Hsp70 from a substrate, promoted by nucleotide exchange, may explain the system's susceptibility to Hsp110. Here, we propose an additional mechanism behind this inhibition. The fact that the NEF and JDP binding occurs at mutually exclusive nucleotide states of Hsp70 has raised a question of competition between the two co-chaperones (*Sousa and Lafer, 2019*), which, to our knowledge, has never been addressed before. On the other hand, class B JDP goes beyond the classical model of the Hsp70 cycle due to the auxiliary interaction with the EEVD motif, which is not nucleotide-sensitive and theoretically might not be exclusive with Hsp110 binding (*Wyszkowski et al., 2021*; *Yu et al., 2015*). Nonetheless, our results clearly demonstrate that Hsp110 disrupts the JDP's complex with Hsp70 (*Figure 4A*, *Figure 4—figure supplement 1C*). This way, Hsp110 may cause the release of Hsp70 from an aggregate, as well as of a JDP, unless it remains tethered via another interaction, e.g., through the other JDP subunit in complex with an aggregate-bound Hsp70 molecule (*Figure 5*). The fate of the displaced proteins is yet to be established, presumably at low Hsp110 levels, they may re-bind at another site on the substrate or to the aggregate-bound chaperones. At high Hsp110 levels, the JDP$^B$-Hsp70 interaction is very strongly affected, similarly as aggregate binding and disaggregation (*Figures 3A, B, D, 4A and B*), suggesting that the proteins released by Hsp110 do not re-bind or that any subsequent binding is non-productive.

The apparent competition between the co-chaperones is reflected in the fact that the inhibition by Hsp110 is moderated at increasing Sis1 level (*Figure 4C*). Interestingly, the class B JDP also exerts a biphasic effect on disaggregation, and optimal concentrations of the two co-chaperones are strongly interdependent (*Figure 4B and C*, *Figure 4—figure supplement 1D and E*). The highest disaggregation yield occurs at Ssa1:Sis1:Sse1 ratio closest to the average reported in the cytosol, 10:0.3:1.7

(*Figure 4C*; *Brownridge et al., 2013*), pointing to a possibility that the proportions between the co-chaperones have undergone evolutionary fine-tuning to develop higher tolerance to stress.

The sub-stoichiometric optimum is unique to Hsp110 (*Gao et al., 2015*; *Wentink et al., 2020*), while other NEF, Fes1, requires 10 times higher level to reach the same degree of stimulation, similarly as the Sse1-2 variant with reduced affinity for Hsp70. Fes1's ability to improve Hsp70 loading onto aggregate is surprising, regarding findings of *Wentink et al., 2020*, who demonstrated, using Hsp110 and BAG1 variants, that Hsp70 clustering on amyloid requires a bulky NEF. Although Fes1 is only 33 kDa, the stretch of its armadillo repeats and different mode of interaction with NBD could possibly lead to similar excluded volume effects as Hsp110. Nevertheless, Fes1 is present in a cell at approximately one-fifth the level of Sse1, and even its 8.4-fold overexpression from two plasmids, shown by Kaimal et al., was insufficient to complement the thermosensitive phenotype of *sse1-200 sse2Δ* (*Kaimal et al., 2017*), as it was six times too low to reach the level effective in disaggregation in vitro (*Figure 1—figure supplement 3A*). It is worth noting that Fes1 has been reported to target substrates to degradation (*Gowda et al., 2018*; *Gowda et al., 2013*) and apparently its role in Hsp70 recruitment to disaggregation is minor.

An abundant association of Hsp70-JDP[B] with aggregate surface, although generating strong pulling effect to disentangle polypeptides, could also mask access of chaperones to the newly emerging sites adjacent to and buried beneath the complex. We demonstrate that Hsp110 could theoretically uncover such sites not only through Hsp70 dissociation from the substrate due to the nucleotide exchange (*Dragovic et al., 2006*), but also by disrupting the JDP[B]-Hsp70 interaction (*Figure 5*). Four Hsp70-binding sites in a JDP[B] dimer allow to form an extensive network of interactions (*Wyszkowski et al., 2021*). We speculate that Hsp110 introduces plasticity into this network, enabling the chaperone complex to infiltrate cavities emerging within the aggregate in a fluid-like manner, binding to the uncovered chaperone-binding sites and pulling up to the aggregate fragmentation (*Figure 5*). Such behaviour would require only a small destabilising effect of the NEF that would not dissociate the complex completely (*Figure 4A*), similarly as observed at the Hsp110 concentration that supports the most effective aggregate binding and disaggregation (*Figure 3A and D*). Further experimental verification of this scenario and future studies of the dynamics within the Hsp70 chaperone system will be critical to understand and combat the stress- and disease-related protein aggregation.

## Materials and methods

### Key resources table

| Reagent type (species) or resource | Designation | Source or reference | Identifiers | Additional information |
|---|---|---|---|---|
| Strain, strain background (*Escherichia coli*) | BL21(DE3) CodonPlus | Agilent | Cat # 230250; RRID:SCR_013575 | Genotype: *E. coli* B F- ompT hsdS(rB- mB-) dcm +Tetr gal endA Hte [argU proL Camr] |
| Strain, strain background (*E. coli*) | RosettaBL21 (DE3) | Novagen | Cat # 70954; RRID:SCR_008441 | Genotype: *E. coli* F- ompT hsdSB(rB- mB-) gal dcm (DE3) pRARE (CamR) |
| Strain, strain background (*Saccharomyces cerevisiae*) | W303 | Laboratory collection | | Genotype: MATa/MATα {leu2-3,112 trp1-1 can1-100 ura3-1 ade2-1 his3-11,15} [phi+] |
| Peptide, recombinant protein | Ssa1 | DOI:10.1073/pnas.0804187105 | Uniprot ID: P10591 | Expressed from pCA533-His6-SUMO-SSA1 plasmid, KanR, T7 |
| Peptide, recombinant protein | Ssa1 ΔEEVD | DOI:10.1016/j.jmb.2015.02.007 | | Expressed from pCA533-His6-SUMO-SSA1 ΔEEVD plasmid, KanR, T7 |
| Peptide, recombinant protein | Sse1 | DOI:10.1073/pnas.0804187105 | Uniprot ID: P32589 | Expressed from pCA534-SSE1 plasmid KanR, T7 |
| Peptide, recombinant protein | Sse1-2 (N572Y N575A) | This study | | Variant generated based on DOI:10.1016/j.jmb.2010.07.004, expressed from pCA534-SSE1-2 plasmid KanR, T7 |
| Peptide, recombinant protein | Fes1 | DOI:10.1073/pnas.0804187105 | Uniprot ID: P38260 | Expressed from pCA707-FES1 plasmid KanR, T7 |
| Peptide, recombinant protein | Ydj1 | DOI:10.1074/jbc.M112.387589 | Uniprot ID: P25491 | Expressed from pET21a-YDJ1 plasmid AmpR, T7 |
| Peptide, recombinant protein | Sis1 | DOI:11810.1073/pnas.2108163118 | Uniprot ID: P25294 | Expressed from pPROEX-TEV-SIS1 plasmid AmpR, T7 |
| Peptide, recombinant protein | Sis1 E50A | DOI:10.1016/j.jmb.2015.02.007 | | Expressed from pPROEX-TEV-SIS1 E50A plasmid AmpR, trc |

*Continued on next page*

*Continued*

| Reagent type (species) or resource | Designation | Source or reference | Identifiers | Additional information |
|---|---|---|---|---|
| Peptide, recombinant protein | Hsp104 | DOI:10.1074/jbc.M112.387589 | Uniprot ID: P31539 | Expressed from pET5a-HSP104 plasmid AmpR, T7 |
| Peptide, recombinant protein | Hsp104 D484K F508A | DOI:10.1016/j.jmb.2019.04.014 | | Expressed from pET5a-HSP104 D484K F508A plasmid AmpR, T7 |
| Peptide, recombinant protein | Hsc70 | DOI:10.1038/nature14884 | Uniprot ID: P11142 | Expressed from pPROEX-His-TEV-HSC70 plasmid AmpR, trc |
| Peptide, recombinant protein | DNAJA2 | DOI:10.1038/nature14884 | Uniprot ID: O60884 | Expressed from pPROEX-His-TEV-DNAJA2 plasmid AmpR, trc |
| Peptide, recombinant protein | DNAJB1 | DOI:10.1038/nature14884 | Uniprot ID: P25685 | Expressed from pPROEX-His-TEV-DNAJB1 plasmid AmpR, trc |
| Peptide, recombinant protein | Hsp105 | DOI:10.1038/nature14884 | Uniprot ID: Q92598 | Expressed from pPROEX-His-TEV-HSP105 plasmid AmpR, trc |
| Peptide, recombinant protein | GFP | DOI:10.1074/jbc.M402405200 | | Expressed from pGFPuv plasmid (TaKaRa, RRID:SCR_003960) |
| Peptide, recombinant protein | Fluc-EGFP | This study | | EGFP fusion with firefly luciferase, expressed from pET22b-Fluc-GFP plasmid AmpR, T7 |
| Peptide, recombinant protein | His-tagged EGFP | This study | | |
| Peptide, recombinant protein | QuantiLum Recombinant Luciferase | Promega | Cat # E1701; RRID:SCR_006724 | |
| Peptide, recombinant protein | His-tagged Luciferase | DOI:10.1371/journal.pgen.1008479 | | |
| Peptide, recombinant protein | Creatine Kinase | Roche | Cat # 10127566001; RRID:SCR_001326 | |
| Commercial assay or kit | Luciferase Assay System | Promega | Cat # E151A; RRID:SCR_006724 | |
| Chemical compound, drug | Alexa Fluor 488 C5 Maleimide | Invitrogen | Cat # A10254; RRID:SCR_013378 | |
| Commercial assay or kit | QuikChange Site-Directed Mutagenesis Kit | Agilent | Cat # 200513; RRID:SCR_013575 | |
| Other | Ni-NTA BLI sensors | Sartorius | Cat # 18-5101; RRID:SCR_003935 | |

## Proteins

Sse1 (*Andréasson et al., 2008a*), Ssa1 (*Andréasson et al., 2008a*), Sis1 (*Wyszkowski et al., 2021*), Ydj1 (*Lipińska et al., 2013*), Hsp104 (*Lipińska et al., 2013*), His-tagged luciferase (*Chamera et al., 2019*), GFP (*Zietkiewicz et al., 2004*) were purified using published protocols. The same protocol as for Sse1 was used for Fes1 (*Andréasson et al., 2008a*). DNAJB4, DNAJA2, Hsc70, and Hsp105 were purified as described in the work by *Nillegoda et al., 2015*. To obtain His-tagged chaperones, a step of proteolytic cleavage of the tag was omitted. Sse1 N572Y N575A was constructed by introduction of point mutations using PCR-specific mutagenesis (Agilent) and confirmed with sequencing. Fluc-EGFP and EGFP (parent vector pCIneo-Fluc-EGFP) were cloned into pET22b plasmid. Fluc-EGFP and His-tagged EGFP were purified using Ni-NTA agarose (Protino), followed by anion exchange chromatography using Q-Sepharose (Q Sepharose Fast Flow, GE Healthcare). Untagged luciferase and its substrate were purchased from Promega (E1702).

## Heat-aggregated luciferase and luciferase-GFP reactivation

Luciferase or luciferase-GFP (30 µM) was denatured in the buffer A (25 mM HEPES-KOH pH 8.0, 75 mM KCl, and 15 mM MgCl$_2$) supplemented with 6 M urea at 25°C for 15 min. Next, it was transferred to 48°C for 10 min and subsequently 25-fold diluted into the buffer A. After 15 min of incubation at 25°C, the reactions were initiated by the addition of the mix of chaperones, which were used at the following concentrations: 1 µM Ssa1, 1 µM Sis1, 1 µM Ydj1, 0.1 µM Sse1, 1 µM Fes1, 3 µM Hsc70, 1 µM DNAJB4, 1 µM DNAJA2, and 0.3 µM Hsp105, if not stated otherwise. Luminescence was measured using Sirius Luminometer (Berthold).

## Unfolded luciferase reactivation

Method for spontaneous folding of luciferase was adapted from *Imamoglu et al., 2020*. Briefly, 10 µM of luciferase was denatured in 5 M GuHCl and 10 mM DTT at 25°C for 1 hr. To initiate the spontaneous folding, the luciferase was 100-fold diluted into the folding buffer (25 mM HEPES-KOH pH 7.5, 100 mM KCl, 10 mM Mg(OAc)$_2$, 2 mM DTT, 0.05% Tween 20). Folding was performed with chaperones used at 1 µM concentration, except for 0.1 µM Sse1, unless it was stated otherwise.

## Renaturation of heat-aggregated GFP

Recovery of GFP aggregates was performed as previously described (*Zietkiewicz et al., 2004*). Briefly, GFP (74 µM) was thermally inactivated at 85°C for 15 min. The reactivation reaction was carried out at 25 °C in the renaturation buffer (25 mM HEPES-KOH [pH 8.0], 7% [vol/vol] glycerol, 60 mM potassium glutamate, 7 mM DTT, 15 mM MgOAc, 10 mM ATP), with an ATP regeneration system comprising 1.2 µM creatine kinase and 20 mM creatine phosphate. Disaggregation of the 100 times diluted GFP aggregates was initiated by adding the chaperone proteins at the following concentrations: Ssa1 (1 µM), Ydj1 (1 µM), Sis1 (1 µM), Sse1 (0.1 µM), Hsp104 WT (1 µM), Hsp104$^{mut}$ (0.15 µM), unless indicated otherwise. GFP fluorescence was detected in a Beckman Coulter DTX880 microplate reader.

## Fluorescent transmitted light microscopy

Luciferase-GFP (14.6 µM) was incubated in the buffer A with 6 M urea at 25°C for 15 min. Then, it was transferred to 48°C for 10 min and 10-fold diluted into the buffer A containing 5 mM ATP and 2 mM DTT. After 15 min of incubation at 25°C, the reaction was initiated upon addition of the mix of chaperones at 1 µM concentration, except for Sse1 used at 0.1 µM. The final concentration of luciferase-GFP aggregates in the reaction was 0.3 µM. After 1 hr of incubation with the chaperones, the reaction was arrested upon addition of 200 mM NaCl and transferred on ice. Specimens were imaged using a confocal laser scanning microscope Leica SP8X with a ×100 oil immersion lens (Leica, Germany). Presented data show results from three independent experiments. Each sample within the repeat was photographed 10 times. Data analysis was performed with Leica LAS X software.

## Dynamic light scattering

Luciferase (23,7 µM) was incubated in the buffer A with 6 M urea at 25°C for 15 min, then it was transferred to 48°C for 10 min. Subsequently, it was 10× diluted with the buffer A containing 5 mM ATP and 1 mM DTT and incubated for 15 min at 25°C. Then, it was subjected to a DLS measurement to confirm the presence of the aggregates, size of which ranged between 1000 nm and 3000 nm. The reaction was initiated upon an addition of a mix of chaperones used at 1 µM concentration, except for Sse1 used at 0.1 µM. The final concentration of luciferase was 0.6 µM. DLS was measured using the Zeta-Sizer NanoS instrument (Malvern) after 1 hr of incubation with chaperones at 25°C. For each sample, three independent measurements were performed, and particle size distributions were calculated as percent within a range between 0.4 nm and 10,000 nm.

## BLI experiments

All the BLI experiments described below were performed as previously described (*Wyszkowski et al., 2021*) using the BLItz and Octet K2 instruments.

## Binding of chaperones to luciferase aggregates

Initially, the Ni-NTA biosensor (ForteBio Dip and Read) was hydrated in the buffer A for 10 min and subsequently immersed in the same buffer containing 6 M urea and 8.2 µM His-tagged luciferase for 10 min, resulting in an anchoring layer of ~6 nm. Next, the biosensor was washed with the buffer A and immersed in the buffer A with 1.6 µM of native His-tagged luciferase and incubated for 10 min at 44°C. Finally, the biosensor was equilibrated with the buffer A supplemented with 5 mM ATP and 2 mM DTT for 5 min. The layer of luciferase aggregates reached ~16 nm and was stable. Binding and dissociation of chaperones were performed in the buffer A supplemented with 5 mM ATP and 2 mM DTT at 25°C.

## Binding of chaperones to the GFP aggregates

To prepare the sensor with GFP aggregates, after the initial hydration of the Ni-NTA biosensor (ForteBio Dip and Read) in the buffer A (25 mM HEPES-KOH pH 8.0, 75 mM KCl, and 15 mM MgCl$_2$)

for 10 min, it was immersed in the buffer A with 9 M urea containing 12.5 µM of His-tagged GFP and incubated at 85°C for 15 min. After washing with the buffer A for 5 min, it was immersed in the buffer A containing 4.2 µM of His-tagged GFP and incubated at 85°C. Next, the sensor was equilibrated with the buffer A supplemented with 5 mM ATP and 2 mM DTT for 5 min. The aggregate layer thickness was ~40 nm. Association and dissociation of chaperones were performed in the same buffer at 25°C.

## Binding of chaperones to the immobilised yeast lysate

Yeast lysate was prepared from overnight culture of W303 yeasts in the YPD medium, according to the published protocol (*Wyszkowski et al., 2021*). The Ni-NTA biosensor (ForteBio Dip and Read) was initially hydrated in the buffer A for 10 min, following incubation in the same buffer supplemented with 6 M urea and 8.2 µM His-tagged luciferase for 10 min, which resulted in biolayer thickness of ~6 nm. Next, the biosensor was washed with the buffer A and immersed in the same buffer containing soluble yeast proteins (5 mg/ml). After 15 min of incubation at 55°C, the biosensor was equilibrated with the buffer A supplemented with 5 mM ATP and 2 mM DTT, reaching biolayer thickness of ~30 nm. Association and dissociation of chaperones were performed in the same buffer at 25°C.

## BLI with fluorescently labelled protein

Ssa1 was incubated with 10× molar excess of Alexa Fluor 488 C5 maleimide (Invitrogen) for 2 hr at 4°C. An excess of the label was removed with a desalting column (PD-10, GE Healthcare). The BLI experiment was performed as with unlabelled proteins and the fluorescence of the dissociated Ssa1 (A488*) was measured using Beckman Coulter DTX 880.

## BLI of direct protein-protein interactions

The biosensor was initially hydrated in the buffer A for 10 min. Next, it was immersed in the buffer A containing the indicated protein (0.4 µM His-Sis1, 2.5 µM His-Sse1, or 1 µM His-Sse1-2) until binding reached saturation, which corresponded to a layer thickness of ~15 nm for His-Sis1, ~6 nm for His-See1 and ~4 nm for His-Sse1-2. Subsequently, it was washed with the buffer A supplemented with 5 mM ATP and 2 mM DTT for 5 min and then immersed in the same buffer with the mix of chaperones at the indicated concentrations.

## Acknowledgements

We thank Bernd Bukau (Heidelberg University), Kevin A Morano (UTHealth Houston), F Ulrich Hartl (Max Planck Institute of Biochemistry), and Claes Andréasson (Stockholm University) for sharing plasmids. We thank Katarzyna Bury, Gabriel Petelski, Maciej Małolepszy, Katarzyna Kalinowska, and Dominik Purzycki for technical support. This work was supported by a grant of the Polish National Science Centre (2019/35/B/NZ1/01475).

## Additional information

### Funding

| Funder | Grant reference number | Author |
|---|---|---|
| Narodowe Centrum Nauki | 2019/35/B/NZ1/01475 | Krzysztof Liberek |

The funders had no role in study design, data collection and interpretation, or the decision to submit the work for publication.

### Author contributions

Wiktoria Sztangierska, Conceptualization, Data curation, Formal analysis, Validation, Investigation, Visualization, Methodology, Writing – original draft; Hubert Wyszkowski, Maria Pokornowska, Conceptualization, Data curation, Formal analysis, Investigation, Methodology, Writing – review and editing; Klaudia Kochanowicz, Investigation, Methodology; Michal Rychłowski, Methodology; Krzysztof Liberek, Conceptualization, Supervision, Funding acquisition, Project administration, Writing – review

and editing; Agnieszka Kłosowska, Conceptualization, Data curation, Formal analysis, Supervision, Investigation, Visualization, Methodology, Writing – original draft, Writing – review and editing

**Author ORCIDs**
Maria Pokornowska (ID) https://orcid.org/0000-0002-3233-7500
Krzysztof Liberek (ID) https://orcid.org/0000-0002-7532-9279
Agnieszka Kłosowska (ID) https://orcid.org/0000-0001-5295-9473

Reviewer #1 (Public review): https://doi.org/10.7554/eLife.94795.3.sa1
Reviewer #2 (Public review): https://doi.org/10.7554/eLife.94795.3.sa2
Reviewer #3 (Public review): https://doi.org/10.7554/eLife.94795.3.sa3
Author response https://doi.org/10.7554/eLife.94795.3.sa4

## Additional files

**Supplementary files**
• MDAR checklist

**Data availability**
All data generated or analysed during this study are included in the manuscript and supporting files; source data files have been provided for Figures 1-4 and all Figure Supplements.

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
