## [Editor Report · eLife assessment]

This study provides an **important** insight into the mechanisms of cooperation between Hsp70 and its cochaperones during reactivation of aggregated proteins. Based on **convincing** evidence, the authors demonstrate that the co-chaperone Hsp110 boosts disaggregation activity by enhancing Hsp70 recruitment to protein aggregates. This work is of broad interest to biochemists and cell biologists working in the protein homeostasis field.

---

## [Referee Report · Reviewer #1 (Public review)]

Summary:

The manuscript by Sztangierska et al explores how the Hsp70 chaperone together with its JDP-NEF cofactors and Hsp104 disentangle aggregated proteins. Specifically, the study provides mechanistic findings that explain what role the NEF class Hsp110 has in protein disaggregation. The results explain several previous observations related to Hsp110 in protein disaggregation. Importantly, the study provides compelling evidence that Hsp110 acts early in the disaggregation process.

Strengths:

(1) This is a very well performed study with multiple in vitro experiments that provide convincing support for the claims.

(2) An important finding is that the study places Hsp110 function early in the disaggregation process.

(3) The study has an important value in that it picks up on a number of observations in the field that have not been explored or directly tested by experiment. The presented results settle questions and controversy regarding Hsp110 function in disaggregation.

Weaknesses:

(1) While the key finding of this manuscript is that it places Hsp110 early in the disaggregation process, the other findings are advancing the field less.

---

## [Referee Report · Reviewer #2 (Public review)]

Sztangierska et al. have investigated the impact of the nucleotide exchange (NEF) factor Hsp110 on the Hsp70-dependent dissolution of amorphous aggregates in the presence of representative members of two classes of J-domain protein.

The authors find that the nucleotide exchange factor of the Hsp110 family, sse1, stimulates the disaggregation activity of yeast Hsp70, ssa1, in particular in the presence of the J-domain protein sis1. Linking chaperone-substrate interactions as determined by biolayer interferometry (BLI) to activity assays, they show that sse1 facilitates the loading of more ssa1 onto the aggregate substrate and propose that this is due to active remodelling of the protein aggregate which exposes more chaperone binding sites and thus facilitates reactivation. This study highlights two important facets of Hsp70 biology: different Hsp70 functions rely on the functional cooperation of specific co-chaperone combinations and the stoichiometry of the different players of the Hsp70 system is an important parameter in tuning Hsp70 chaperone activity.

Strengths:

The manuscript presents a systematic analysis of the functional cooperation of sse1 with a class B J-domain protein sis1 in the disaggregation of two different model aggregate substrates, allowing the authors to draw more general conclusions about Hsp70 disaggregation activity.

The authors can pinpoint the role of sse1 to the initial remodeling of aggregates, rather than the later stages of refolding, highlighting the functional specificity of Hsp70 co-chaperones.

They demonstrate the competitive nature of binding to ssa1 between sse1 and sis1 which can explain the poisoning of Hsp70 chaperone activities observed at high NEF concentrations.

Weaknesses:

While structural requirements have been identified that allow sse1, in cooperation with sis1, to facilitates the loading of Hsp70 on the amorphous aggregate substrate, how this is achieved on a mechanistic level remains an open question.

---

## [Referee Report · Reviewer #3 (Public review)]

Summary:

The authors studied the function of Hsp110 co-chaperones (e.g. yeast Sse1) in Hsp70-dependent protein disaggregation reactions. The study builds on former work by the authors (Wyszkowski et al., 2021, PNAS), analyzing the binding of Hsp70 and J-domain protein (JDP) cochaperones to protein aggregates using bio-layer interferometry (BLI). It was shown before by other groups that Hsp110 enhances Hsp70 disaggregation activity. The mechanism of Hsp110-stimulated disaggregation activity, however, remained poorly defined. Here, the authors show that yeast Hsp110 increases Hsp70 recruitment to the surface of protein aggregates. The effect is largely dependent on J-domain protein (JDP) identity and particularly pronounced for class B JDPs (e.g. yeast Sis1), which are also more effective in disaggregation reactions. The authors also confirm former results, showing inhibition by increased Hsp110 levels and provide novel evidence that the inhibitory effect is caused by competition between Hsp110 and JDPs for Hsp70 binding.

Strengths:

The work represents a very thoroughly executed study, which provides novel insights into the mechanism of Hsp70-mediated protein disaggregation. Key findings established for yeast chaperones are also documented for human counterparts. The observation that Hsp110 might displace JDPs from Hsp70 during the disaggregation reaction is very appealing. It will now become important to validate this initial finding and dissect how it propels the disaggregation reaction.

Weaknesses:

How exactly the interplay between JDPs and Hsp110 orchestrates protein disaggregation remains largely speculative and further analysis is required for a deeper mechanistic understanding.

---

## [Author Response]

The following is the authors’ response to the original reviews.

**Reviewer #1 (Recommendations For The Authors):**
- The title may not reflect the key finding of the paper. It is well established in the field that the disaggregation process is sensitive to perturbations of the levels of the disaggregating factors.

We have changed the title to better reflect the major finding of the work, the importance of the NEF during the initiation of disaggregation. The new title is: *Early Steps of Protein Disaggregation by Hsp70 Chaperone and Class B J-Domain Proteins are Shaped by Hsp110*.

- Abstract:Please note that the phrases "stimulation is much limited with class A JDPs", "limited destabilization of the chaperone complex improves disaggregation", and "tuned proportion between the co-chaperones" are hard to understand. Only after having read the manuscript are the meanings of these phrases accessible.

The phrases in the abstract were changed (page 1, lines 10-14).

- The subheading "Sse1 improves aggregate modification by Hsp70" on p. 7 is unclear. What is measured is a decrease in aggregate size dependent on Hsp70-JDP as well as Sse1.

The subheading was changed to include more precise information, into “Sse1 leads to Hsp70-depenent reduction of aggregate size”.

- The subheading "Biphasic effects of Sse1 on the Hsp70 disaggregation activity" does not describe the finding clearly; "Biphasic effects" is a term that is hard to understand.

To avoid phrases that can be understood in many ways, we have changed the subheading into “Hormetic effects of Sse1 in Hsp70 disaggregation activity”

- p.5, last line. Hsp110 typo The typos have been corrected.
**Reviewer #2 (Recommendations For The Authors):**
(1) The article emphasises multiple times the importance of stoichiometry between the (co-)chaperones. Most figures would benefit from an indication of the used stoichiometry (or all absolute concentrations) to support the points made about the stoichiometry, especially the figures showing titrations of Sse1, Sse1-2, and Sis1 (Fig. 3D, 3E, 4A-C, S2B, S5F, S6A-E).

The information of protein concentrations has been included in all figure captions.

(2) The manuscript includes a summary model. While this model is a plausible hypothesis of the mechanism of disaggregation by Hsp70, in particular when viewed with previous data (Wyszkowski et al., 2021), it focuses rather heavily on the potential remodeling of clients by Hsp70, which is not the primary focus of the data presented in this manuscript. More emphasis could be put on the JDP class/ functional specificity observed.

The model has been changed according to the Reviewer’s comments to better reflect the findings presented in the manuscript (Figure 5).

(3) The methods section is very brief. I recommend including additional details about reaction conditions (temperature, buffer compositions, protein concentrations) even when previously reported elsewhere to improve the readability of the manuscript. Details regarding the DLS experiments performed are missing.

More detailed information on the experimental conditions has been added to the Methods section, as well as to figure legends.

(4) Many experiments incorporate BLI to assess the effect of NEFs on the binding of the Hsp70 and JDP to aggregates. Although appropriate controls are included (no ATP, Hsp70, and JDP only), a control with only Hsp70 and the NEF would be useful to determine to which extent the NEF itself alters the thickness of the (Hsp70-bound) aggregate biolayer.

The suggested controls were added (Figure 1—figure supplement 1 G) and discussed in the manuscript (page 5, lines 23-24).

**Reviewer #3 (Recommendations For The Authors):**
- The refolding assay makes use of Luciferase denatured in 5 M GdnHCl. These conditions lead to a spontaneous refolding yield of 20% (Figure 3C), which is very high and limits conclusions on the effect of Hsp110 but also JDPs on the refolding process. Typically this assay uses 6 M GdnHCl for Luciferase denaturation and under these conditions, spontaneous refolding of Luciferase is hardly observed (e.g. Laufen et al. PNAS 1999). The authors are therefore asked to repeat key experiments using altered (6M) GdnHCl concentrations.

We based our experiments assessing luciferase refolding on the publication by Imamoglu et al. (2020), in which the authors, using 5 M GdnHCl for luciferase denaturation, demonstrated that spontaneous and chaperone-assisted luciferase refolding strongly depends on luciferase concentration. In this work, a similar degree of luciferase refolding was reported for the same final luciferase concentration (100 nM) as we used in our experiments (Figure 1—figure supplement 1D). As an additional control, we compared the effects of 5 M and 6 M of GdnHCl during denaturation on luciferase refolding under the same conditions (100 nM, 25 °C, 2 h) and we observed no significant differences (Author response image 1).

**Author response image 1. sa4fig1:** Chaperone-assisted folding of luciferase after denaturation at 5 M or 6 M GdnHCl. Luciferase was denatured in 5 M or 6 M GdnHCl according to the protocol in the Materials and Methods section. Luminescence was monitored alone or after incubation with Luminescence was monitored alone or after incubation with Ssa1-Sis1 or Ssa1-Ydj1. Chaperones were used at 1 µM concentration. Luciferase activity was measured after 2 hours and normalized to the activity of the native protein. Error bars indicate SD from three repeats.

- Figure 1B: The authors are asked to provide binding curves for Ssa1/Sse1 (no Sis1) and Sis1/Sse1 (no Ssa1) as controls. Particularly the latter combination is required as direct cooperation between Hsp110 and JDPs has been suggested in the literature (Mattoo et al., JBC 2013).

We performed the suggested BLI experiment, and the results are presented in the new Figure 1—figure supplement 1 G (page 5, lines 23-24).

- Figure 1B (and other figure parts showing BLI data): it is unclear how often the BLI experiments have been performed. This should be stated in the figure legend. Can the authors add SDs to the respective curves?

We added detailed information about the number of replicates to the figure legends. SD bars were added to the BLI results shown in Figures1-4, apart from the results of titrations, for which, for the sake of clarity, the three replicates are represented in the plots on the right (Figure 3D). In the case of less than 3 repeats of the results presented in the Supplementary Figures, the remaining repeats are added to the provided Source Data file, information about which has been added to the captions of the respective figures.

- The observation that Hsp110 can interrupt Hsp70 interaction with JDPs is intriguing. Do the authors envision JDP displacement from the aggregate? If so this could be shown in BLI experiments by monitoring the release of fluorescently labeled Sis1 (similar to labeled Ssa1, Fig. S3C). Or will the released JDP immediately rebind to another binding site on the aggregate? The authors should at least discuss the diverse scenarios as they are relevant to the mechanism of protein disaggregation.

The proposed experiment is challenging due to the transient nature of Sis1 binding to aggregate and high background observed with the method using the fluorescently labelled proteins. The aspect of chaperone’s re-binding after their release by Hsp110 proposed by the reviewer has been introduced into the Discussion section (pages 12/13, lines 25-4). We speculate that Hsp110 might release an Hsp70 molecule as well as a JDP molecule that had been bound to the aggregate through Hsp70 (Figure 5).

- Figure 2B: Ssa1/Sis1/Sse1 strongly decreases the size of Luciferase-GFP aggregates. Yet this activity only allows for limited refolding of aggregated Luciferase and the reaction stays largely dependent on Hsp104. How do the authors envision the role of the hexameric disaggregase in this process? Does it act exclusively on small-sized aggregates after Hsp110-dependent fragmentation?

A question of the Hsp104 activity with the Hsp70-processed aggregates is indeed intriguing and we agree that it should have been discussed more thoroughly. We added to the manuscript the results of the reactivation of luciferase-GFP with and without Hsp104 to emphasize the role of Hsp104 in the active protein recovery (Figure 2—figure supplement 1A) (page 7, lines 24-27). We propose that aggregate fragmentation by Hsp70-JDPB-Hsp110 increases the effective aggregate surface, at which Hsp104 might become engaged. We do not think that Hsp104 acts only on small aggregates, it might be just more effective, when the number of exposed polypeptides is larger. In the cell, where Hsp104 binds to aggregates of various sizes, protein aggregates apparently also need to undergo such Hsp110-boosted pre-processing by Hsp70, based on the finding that Sse1 is not necessary for Hsp104 recruitment to aggregates, but it is required for Hsp104-dependent disaggregation (Kaimal et al., 2017). We have added a comment on this problem to the Discussion section (pages 11/12, lines 33-4) .

- Page 9: The authors state that the Sse1-2 variant is nearly as effective as Sse1 Wt in stimulating substrate dissociation and refer to published work (Polier et al., 2008). It is unclear how the variant should have Wtlike activity in triggering substrate release although its activity in catalyzing nucleotide exchange is reduced to 5% (both activities are coupled). The observation that high Sse1-2 concentrations do not inhibit protein disaggregation does not necessarily exclude the possibility that high Sse1 WT concentration inhibit the reaction by overstimulating substrate release. The latter possibility should be considered by the authors and added to the discussion section.

We agree with the Reviewer that the description of the Sse1-2 variant was misleading, as it was lacking the key information, that according to the published data (Polier et al., 2008), it was 10 times higher the concentration of the Sse1-2 variant than Sse1 WT that had a similar nucleotide-exchange activity to the wild type. We have changed the text (page 9, lines 16-22, page 13, lines 26-28) to avoid confusion as well as the model in the Figure 5, to underline the importance of substrate release as the cause of the Hsp110-dependent inhibition.

- While similar effects are observed for human class A and class B JDP co-chaperones, they are clearly less pronounced. A mechanistic explanation for the difference between yeast and human chaperones is currently missing and the authors are asked to elaborate on this aspect.

There are indeed clear differences between the human and yeasts systems, especially regarding the dependence on the NEF. Hsc70 has been reported to have a lower rate of ADP release (Dragovic et al., 2006) and thus might rely more on Hsp110 than its yeast ortholog. For the same reason, the strong Hsc70 stimulation by Hsp105 is also observed with class A JDP. We have added a comment on these effects in the Discussion section (page 12, lines 17-21).

Minor points- Figure S1C (right): the disaggregation rate (%GFP/h) is somewhat misleading/confusing as a value of more than 150%/h is determined in the presence of the complete disaggregation system while only approx. 60% GFP is indeed refolded by the system (Figure S1C, left). Showing the rate as %GFP/min seems more rational.

We changed the units according to the Reviewer’s comment (Figure 1—figure supplement 1A, C).

- Figure S5B: Only a single data point is shown for Ssa1/Sis1/Sse1.

We changed the figure to include datapoints from all three repeats (Figure 3—figure supplement 1 B).

- There are several typos throughout the manuscript. A more careful proofreading is recommended

We have corrected the typos.

**Reviewer #1 (Public Review):**
The experiments differ somewhat in regard to the aggregated protein used. For example, in Figure 1A, FFL is used with only limited reactivation (10% reactivated at the last timepoint and the curve is flattening), while in Figure 2B FFL-EGFP is used to monitor microscopically what appears to be complete disaggregation. Does FFL-EGFP behave the same as FFL in assays such as the one in Figure 1A or are there major differences that may impact how the data should be interpreted?

We added the results of Luc-GFP reactivation (Figure 2—figure supplement 1 B) (discussed on page 7, lines 24-27 of the manuscipt) which agree with the results obtain with Luciferase as a substrate (Figure 1—figure supplement 1 B). They clearly show that the Ssa1-Sis1-Sse1-dependent decrease in aggregate size is not associated with the recovery of active protein.

**Reviewer #2 (Public Review):**
Experimental data concerning the class A JDPs should be interpreted with caution. These experiments show very small reactivation activities for luciferase in the range of 0-1% without the addition of Hsp104 and 0-15% with the addition of Hsp104. Moreover, since the assay is based on the recovery of luciferase activity, it conflates two chaperone activities, namely disaggregation and refolding. It is possible that the small degree of reactivation observed for the class A JDP reflects a minor subpopulation of the aggregated species that is particularly easy to disaggregate/refold and may thus not be representative of bulk behaviour.

The disaggregation by the Hsp70 system can be enhanced by the addition of small heat shock proteins at the step of substrate aggregation (Rampelt et al., 2012). However, sHsps compete with Hsp70 for binding to the aggregate (Żwirowski et al., 2017) and for that reason we decided not to include sHsps in the experiments presented in the manuscript, as it would introduce another level of complexity. However, as a control, we performed the disaggregation assay with Hsp70 with Ydj1 using luciferase aggregates formed in the presence or absence of sHsp (Author response image 2). In 1 h, the Hsp70 system without Hsp104 yielded 5% of recovered luciferase activity and the system with Hsp104, 23% compared to the native. The impact of Sse1 on Ssa1-Ydj1 and Ssa1-Ydj1-Hsp104 was similar as for luciferase aggregates formed without sHsps (Figure 1A, Figure 1—figure supplement 1 B). Furthermore, according to the Reviewer’s comment, we have changed the Figure 5 to underscore the more prominent role of class A JDPs in the final protein folding than in disaggregation.

**Author response image 2. sa4fig2:** Disaggregaton of heat-aggregated luciferase – impact of sHsps. Luciferase (2 μM) was denatured with (blue) or without (red) Hsp26 (20 μM) at 45 °C for 15 min in the buffer A (Materials and Methods). Upon 100-fold dilution with the buffer A, supplemented wih 5 mM ATP, 2 mM DTT, 1.2 μM creatine kinase, 20 mM creatine phosphate, chaperones indicated in the legend were added to the final concentration of 1 μM, except for Sse1, concentration of which was 0.1 μM. Shown is luciferase activity measured after 1 h of incubation at 25 °C, normalized to the activity of native luciferase.

**Reviewer #3 (Public Review):**
Enhanced recruitment of Hsp70 in the presence of Hsp110 was shown for amyloid fibrils before (Beton et al., EMBO J 2022) and should be acknowledged.

We have added the suggested citation with a respective comment (page 11, lines 20-21).